# Real-Time Urethral and Ureteral Assessment during Radical Cystectomy Using Ex-Vivo Optical Imaging: A Novel Technique for the Evaluation of Fresh Unfixed Surgical Margins

Francesco Prata [1,*], Umberto Anceschi [2], Chiara Taffon [3], Silvia Maria Rossi [3], Martina Verri [4], Andrea Iannuzzi [1], Alberto Ragusa [1], Francesco Esperto [1], Salvatore Mario Prata [5], Anna Crescenzi [3], Roberto Mario Scarpa [1], Giuseppe Simone [2] and Rocco Papalia [1]

1   Department of Urology, Fondazione Policlinico Universitario Campus Bio-Medico, 00128 Rome, Italy
2   Department of Urology, IRCCS "Regina Elena" National Cancer Institute, 00144 Rome, Italy
3   Pathology Unit, Fondazione Policlinico Universitario Campus Bio-Medico, 00128 Rome, Italy
4   Unit of Endocrine Organs and Neuromuscular Pathology, Fondazione Policlinico Universitario Campus Bio-Medico, 00128 Rome, Italy
5   Simple Operating Unit of Lower Urinary Tract Surgery, SS. Trinità Hospital, 03039 Sora, Italy
*   Correspondence: f.prata@unicampus.it; Tel.: +39-3934373027; Fax: +39-06225411995

**Abstract:** Background: Our study aims to assess the feasibility and the reproducibility of fluorescent confocal microscopy (FCM) real-time assessment of urethral and ureteral margins during open radical cystectomy (ORC) for bladder cancer (BCa). Methods: From May 2020 to January 2022, 46 patients underwent ORC with intraoperative FCM evaluation. Each specimen was intraoperatively stained for histopathological analysis using FCM, analyzed as a frozen section (FSA), and sent for traditional H&E examination. Sensitivity, specificity, positive predictive value (PPV), and the negative predictive value (NPV) of FCM and FSA were assessed and compared with H&E for urethral and ureteral margins separately. Results: The agreement was evaluated through Cohen's κ statistic. Urethral diagnostic agreement between FCM and FSA showed a κ = 0.776 ($p < 0.001$), while between FCM and H&E, the agreement was κ = 0.691 ($p < 0.001$). With regard to ureteral margins, an overall agreement of κ = 0.712 ($p < 0.001$) between FCM and FSA and of κ = 0.481 ($p < 0.001$) between FCM and H&E was found. Conclusions: FCM proved to be a safe, feasible, and reproducible method for the intraoperative assessment of urethral and ureteral margins during ORC. Compared to standard FSA, FCM showed adequate diagnostic performance in detecting urethral and ureteral malignant involvement.

**Keywords:** bladder cancer; intraoperative analysis; fluorescent confocal microscopy; radical cystectomy; surgical margins





## 1. Introduction

Radical cystectomy (RC) with pelvic lymphadenectomy represents the cornerstone of management for muscle-invasive bladder cancer (MIBC) [1,2]. Indications for radical treatment also range from BCG-resistant/recurrent non-muscle invasive high-grade bladder cancer (NMIBC) [3] to palliative intent for locally advanced disease determining the choice of urinary diversion. Due to the panurothelial nature of bladder cancer (BCa), urethral and ureteral recurrences are not uncommon after RC accounting for an incidence ranging from 1–8% and 4–10%, respectively [4–6]. Regardless of the surgical approach considered, in the next 5 years, more than 50% of patients treated by RC will experience a recurrence of urothelial carcinoma (UC) or eventually die from progression to metastatic disease [7–9] with major consequences on the patient's health and prognosis, underlying the aggressiveness and recurrence of BCa.

Despite evidence supporting the impact of positive surgical margins (PSM) on recurrence and survival, guidelines do not provide a clear indication of the necessity of

performing frozen section analysis (FSA) for ureteral margins during RC [1,10]. Moreover, FSA's ability to detect BCa intraoperatively and its prognostic role remains controversial as the lack of agreement with final pathology examination and patients' recurrence. A recent systematic review reported FSA of ureteral margins to have sensitivity and specificity ranging between 69–77% and 83–96%, respectively, while for urethral margins, the sensitivity varied from 33% to 93% and specificity from 99% to 100% [11]. This variability of FSA, in its unclear prognostic role, points out the need for a more reliable standardized method for the intra-operative assessment of surgical margins infiltration.

Fluorescence confocal microscopy (FCM) represents an optical technique that allows the immediate acquisition of digital images from ex vivo fresh tissue. The main advantage of FCM analysis is represented by preserving tissue integrity for conventional histopathological and immunohistochemical analysis. This novel technology has been recently applied in the uro-oncology field for detecting prostate cancer at surgical margins during robot-assisted radical prostatectomy with excellent results [12–14], while its role in the RC scenario has never been explored.

The aim of this study was to assess the diagnostic accuracy of FCM for ureteral and urethral margins during RC and to compare the outcomes with those obtained with standard FSA on a contemporary series of ORCs performed at a single center.

## 2. Materials and Methods

### 2.1. Patients and Dataset

Our institutional, board-approved BCa dataset was queried for "open radical cystectomy" and "FCM". Between May 2020 and January 2022, all patients who were scheduled for ORC for primary BCa (including squamous cell carcinoma) for either curative or palliative intent were enrolled (*n* = 62). Since 2016, it has been a routine practice at our center to perform FSA of the urethra and distal ureters at the time of RC. Exclusion criteria were as follows: patient < 18 years old (*n* = 0); RC performed for non-primary BCa reasons (i.e., neurologic bladder, chronic cystitis, other primary tumor invasion; *n* = 7); RC plus concomitant nephroureterectomy for upper urinary tract urothelial cell carcinoma (UTUC) (*n* = 5); previous nephroureterectomy for UTUC (*n* = 4). A total of 46 eligible patients were identified for analysis. Baseline data collected included gender, age, BMI, major comorbidities, preoperative pathological stage, genitourinary instillations, preoperative hydronephrosis (unilateral/bilateral), and % neoadjuvant chemotherapy (NAC). Perioperative data included median operative time, type of urinary diversion (continent/incontinent), tumor stage, histology and grading, surgical margin status, median lymph node yield, and urethral and ureteral margins. According to international guidelines, the patient's follow-up timeframe was adapted to the pT stage [1].

Positive margins were defined as the presence of UC and CIS at initial FCM evaluation and confirmation by FSA specimen at the final pathological examination, while dysplasia or atypia were excluded from the analysis as possible confounding variables. If positive margins were identified in the urethra, the patient did not receive an orthotopic urinary bladder reconstruction, and simultaneous urethrectomy was performed in all cases. If the ureteral margin was positive or suspicious for cancer infiltration, an additional ureter resection was performed cranially and sent for intraoperative evaluation until a negative margin status was achieved at FSA.

The primary endpoint of the analysis was to assess the level of agreement between the intraoperative FCM and FSA of urethral and ureteral margins. Secondary endpoints were as follows:

- to estimate the level of agreement between FCM and final histological diagnosis;
- to assess the reliability of this technique for evaluating cancer infiltration on surgical specimens (ureters/urethra) during ORC.

### 2.2. Fluorescent Confocal Microscopy

FCM analysis was performed using VivaScope® 2500 (MAVIG, Munich, Germany). This confocal laser-scanning microscope combines two lasers of different wavelengths (a 488 nm blue laser for fluorescence signal and a 638 nm infrared laser for reflection signal) to create two separated images: a fluorescence one and a corresponding reflectance image, respectively. A built-in device algorithm translates the acquired image information from both lasers' signals into pseudo-colored images that resemble the H&E stain of conventional histological sections. VivaScope® enables tissue examination with a vertical resolution of approximately 5 μm and a maximum examination depth of 200 μm. The reconstructed image represents a collection of mosaic images with a maximum total scan area of $25 \times 25$ mm and a maximum image resolution of $51,000 \times 51,000$ pixels.

### 2.3. Tissue Preparation

Each specimen was sent to the pathology department and treated with a drop of acridine orange (0.6 mM; Sigma-Aldrich®, St. Louis, MO, USA) for at least one minute. Then, it was fully washed with a sterile saline solution of 0.9% to avoid specimen contamination from any acridine orange (Figure 1), drained on absorbent paper, and placed between two glass slides in a sandwich manner. Slides were allocated into the VivaScope® 2500 slot for real-time analysis (Figure 2). FCM images were recorded in a dedicated hard disk along with the patient's identification number, intervention date, and pathological evaluation. Tissue preparation, image acquisition, and final evaluation required less than 5 minutes for each case. After this processing, the same fresh unfixed specimens were removed from Vivascope and sent to standard FSA. Frozen sections were obtained by freezing the specimens, cutting within a cryostate, and staining with H&E. Finally, samples were defrosted and embedded in FFPE for final histological evaluation.

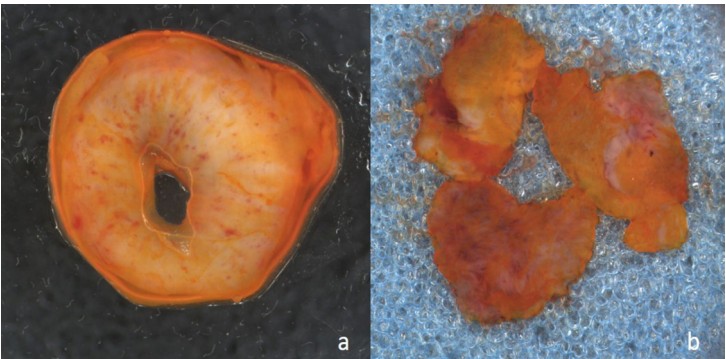

**Figure 1.** Macroscopic image of resection margin prepared with acridine orange for Vivascope analysis: (**a**) from a ureter, and (**b**) from the urethra.

### 2.4. Statistical Analysis

Sensitivity, specificity, positive predictive value (PPV), and negative predictive value (NPV) of FCM and FSA were assessed and compared with definitive histological diagnoses (formalin fixed paraffin embedded, FFPE) for urethral and ureteral margins separately. The agreement rates between FCM and FSA or FFPE diagnoses were evaluated through Cohen's κ statistic [15]. Interpretation of agreement was described as follows: no agreement (κ = 0), slight agreement (κ = 0.01–0.20), fair agreement (κ = 0.21–0.40), moderate agreement (κ = 0.41–0.60), substantial agreement (κ = 0.61–0.80), and almost total or total agreement (κ = 0.81–0.99). Furthermore, the interpretation of reproducibility was evaluated as: marginal (κ < 0.40), good (κ = 0.40–0.75), and excellent (κ > 0.75). For all analyses, a $p < 0.05$ was considered statistically significant. IBM Statistical Package for the Social Sciences (SPSS) statistical software package (IBM Corp. Released 2020. IBM SPSS Statistics for Windows, Version 27.0. Armonk, NY, USA: IBM Corp) was used for statistical analyses.

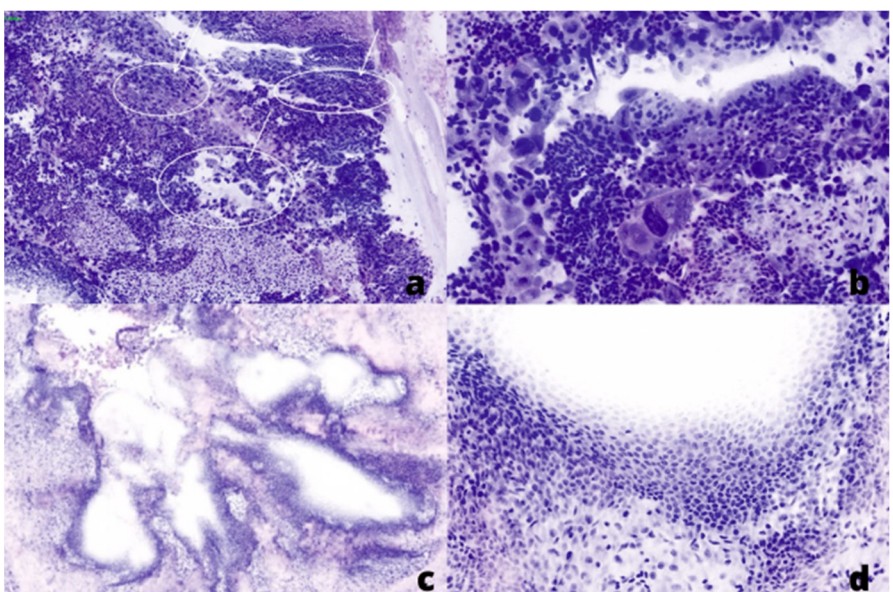

**Figure 2.** Fluorescent (FCM) digital microscopic images showed in Hematoxylin and Eosin (H&E) pseudo-colors: (**a,b**) from a positive ureter margin with high-grade urothelial cell carcinoma, low (×20) and high (×40) power field, respectively. (**c,d**) from a negative ureter margin, low (×5) and high (×10) power field, respectively.

## 3. Results

Overall, 46 patients underwent ORC with intraoperative FCM control of urethral and ureteral margins at our institution in the timeframe considered. A total of 138 specimens were analyzed with FCM: 46 urethral, 46 left ureteral, and 46 right ureteral margins. Clinical and baseline characteristics were reported in Table 1. Only 6 patients (13%) underwent NAC due to either locally advanced BCa (cT > 2b) or cN+ at the time of diagnosis with gemcitabine-cisplatin (GC) as the drug combination regimen in all cases. Concomitant CIS to UC was reported preoperatively in five patients (10.9%), while 11 patients (23.9%) revealed CIS presence at final pathology. Compared to FSA, FCM reported a prevalence of positive surgical margins of 5/5 (100%) and 7/9 (77.8%) for urethral and ureteral specimens, respectively. While compared to final histology, FCM reported a prevalence of 5/6 (83.3%) and 7/9 (77.8%) positive surgical margins for urethral and ureteral specimens, respectively. Histology type at final pathology showed that the majority of patients had an urothelial carcinoma (43/46, 93.5%), while only two patients displayed other variants of BCa (1/46 squamous carcinoma and 1/46 cribriform variant). After a median follow-up of 11.5 (5–15) months, only one patient (2.2%) had a recurrence of urothelial cancer of the upper urinary tract.

**Table 1.** Clinical and baseline data.

| Variable | Results |
|---|---|
| Male ($n$, %) | 37 (80.4%) |
| Female ($n$, %) | 9 (19.6%) |
| Age (yrs, median) | 74 (70–79) |
| BMI (kg/m$^2$, median) | 25.36 (22.4–28.1) |
| ASA score ($n$, %)<br>    1–2<br>    3–4 | <br>20 (43.5%)<br>26 (56.5%) |

**Table 1.** *Cont.*

| Variable | Results |
| --- | --- |
| Smoking History | 35 (76.1%) |
|   Current Smoker (*n*, %) | 16 (34.8%) |
|   Former Smoker (*n*, %) | 19 (41.3%) |
|   No smoking history (*n*, %) | 11 (23.9%) |
| Diabetes (*n*, %) | 10 (21.7%) |
| Hypertension (*n*, %) | 33 (71.7%) |
| BCG irrigations (*n*, %) | 6 (13%) |
| Neoadjuvant chemotherapy (*n*, %) | |
|   GC (*n*, %) | 6 (13%) |
|   Other regimens (*n*, %) | 0 (0%) |
| Neoadjuvant radiotherapy (*n*, %) | 0 (0%) |
| Preoperative Hydronephrosis (*n*, %) | 24 (52.2%) |
|   Left | 15 (32.6%) |
|   Right | 16 (34.8%) |
|   Bilateral | 7 (15.2%) |
| Preoperative CKD stage (*n*, %) | |
|   1 | 7 (15.2%) |
|   2 | 18 (39.1%) |
|   3a | 10 (21.7%) |
|   3b | 6 (13%) |
|   4–5 | 5 (10.9%) |
| cT (*n*, %) | |
|   1 | 8 (17.4%) |
|   2 | 25 (54.3%) |
|   3 | 9 (19.5%) |
|   4 | 4 (8.7%) |
| Cis (*n*, %) | 5 (10.9%) |
| cN (*n*, %) | 9 (19.5%) |
|   1 | 0 (0%) |
|   2 | 7 (15.2%) |
|   3 | 2 (4.3%) |
| Incidental PCa (*n*, %) | 13 (351%) |
| Urinary diversion (*n*, %) | |
|   Continent | 11 (23.9%) |
|   Incontinent | 35 (76.1%) |
| pT (*n*, %) | |
|   0 | 4 (8.7%) |
|   1 | 11 (23.9%) |
|   2a | 2 (4.3%) |
|   2b | 7 (15.2%) |
|   3a | 14 (30.4%) |
|   3b | 1 (2.2%) |
|   4a | 6 (13%) |
|   4b | 1 (2.2%) |
| pTis (*n*, %) | 11 (23.9%) |
| pN (*n*, %) | 11 (23.9%) |
|   1 | 3 (6.5%) |
|   2 | 7 (15.2%) |
|   3 | 1 (2.2%) |

**Table 1.** *Cont.*

| Variable | Results |
|---|---|
| Histology (type, *n*, %) | |
|    Urothelial | 43 (93.5%) |
|    Squamous | 1 (2.2%) |
|    Cribriform | 1 (2.2%) |
| OS (*n*, %) | 25 (54.3%) |
| CCS (*n*, %) | 45 (97.8%) |
| MFS (*n*, %) | 33 (71.7%) |
| Recurrence | |
|    Upper urinary tract | 1 (2.2%) |
|    Urethral | 0 (0%) |
|    Local | 0 (0%) |
| Follow-up (months) (median, IQR) | 11.5 (5–15) |

Abbreviations: BMI: body mass index; ASA score: American Society of Anesthesiologists score; BCG: Bacillus Calmette-Guérin; GC: Gemcitabine-cisplatin; CKD; chronic kidney disease; PCa: Prostate Cancer; OS: overall survival; CSS: cancer-specific survival; MFS: metastases-free survival.

FCM, compared to FSA, showed an 80% sensitivity, 97.8% specificity, 80% PPV, and 97.6% NPV for urethral margins, respectively; 66.7% sensitivity, 97.5% specificity, 80% PPV, and 95.1% NPV were observed when compared to H&E (Table 2).

**Table 2.** Concordance evaluation of FCM at a urethral level between FSA and final histology.

| Variable | FSA | Final Histology |
|---|---|---|
| Prevalence | 5/5 (100%) | 5/6 (83.3%) |
| Sensitivity | 80% | 66.7% |
| Specificity | 97.8% | 97.5% |
| Positive Predictive Value (PPV) | 80% | 80% |
| Negative Predictive Value (NPV) | 97.6% | 95.1% |
| K Cohen | 0.776 (*p* < 0.001) | 0.691 (*p* < 0.001) |

Abbreviations: FCM: fluorescent confocal microscopy; FSA: frozen section analysis.

Concerning ureteral margins, FCM, compared to FSA, showed a 69.2% sensitivity, 97% specificity, 90% PPV, and 88.9% NPV; while a 53.8% sensitivity, 90.9% specificity, 70% PPV, and 83.3% NPV were evidenced when compared to H&E (Table 3). The FCM false positive rate of overall adverse pathology was relatively low (8.9%).

**Table 3.** Concordance evaluation of FCM at a ureteral level between FSA and final histology.

| Variable | FSA | Final Histology |
|---|---|---|
| Prevalence | 7/9 (77.8%) | 7/9 (77.8%) |
| Sensitivity | 69.2% | 53.8% |
| Specificity | 97% | 90.9% |
| Positive Predictive Value (PPV) | 90% | 70% |
| Negative Predictive Value (NPV) | 88.9% | 83.3% |
| K Cohen | 0.712 (*p* < 0.001) | 0.481 (*p* < 0.001) |

Urethral diagnostic agreement between FCM and FSA analysis showed a κ = 0.776 (*p* < 0.001), while between FCM and H&E, the agreement was κ = 0.691 (*p* < 0.001). With regard to ureteral margins, an overall agreement between FCM and FSA of κ = 0.712 (*p* < 0.001) was evidenced, and of κ = 0.481 (*p* < 0.001) comparing FCM to FFPE. The interpretation of the agreement was substantial in almost all cases except for FCM and H&E for ureters (κ = 0.481). On the other hand, the interpretation of reproducibility between FCM and FSA was acceptable for almost all cases, and excellent for the urethral margin (κ = 0.776).

## 4. Discussion

In the RC setting, the best intra-operative assessment methodology for evaluating urethral and ureteral margins is still far from being standardized. The prognostic role of FSA remains uncertain, to the point that current international guidelines do not recommend performing an intra-operative evaluation of surgical margins [1]. Despite the evidence of performing urethral and ureteral sections intraoperatively, histopathological examination with traditional FSA is based on tissue freezing and cutting, followed by staining glass slides and its analysis by a local pathologist using a conventional microscope. This is often time-consuming, requires human and technological resources on site, and may cause loss or distortion of the fresh tissue. Moreover, the process for the acquisition of FSA glass slides could sometimes lead to the carcinoma being cleared from sections during sampling and consequently to a negative margin at final pathology, with a substantial number of false negatives. This is why we avoided a direct comparison between FCM/FSA and H&E. The controversial role of FSA is supported by the lack of strong guideline recommendations. In this context, a technological solution is strongly required that speeds up intraoperative evaluation and avoids the deterioration of the biological sample for the definitive diagnosis. According to current literature, the present study represents the first series attempting a head-to-head comparison of FCM vs. FSA and FCM vs. FFPE during ORC and trying to explore the role of this new methodology in the bladder cancer scenario. Among novel optical technologies, FCM is gaining attention and space. This optical imaging modality uses the inherent light-scattering properties of the different components of the tissue generating optical sections of biological specimens, like H&E-stained tissue sections obtained by conventional cutting frozen or fixed tissues. FCM can generate optical sections of biological specimens allowing real-time microscopic evaluation of fresh unfixed tissue, requiring minimal tissue preparation and preserving specimens from damage, distortion, or loss.

Our results showed a high diagnostic agreement of urethral margin for FCM when compared to FSA and H&E FFPE (κ = 0.776, *p* < 0.001, and κ = 0.691, *p* < 0.001, respectively) while, concerning ureteral margins, overall diagnostic ureteral agreement for FCM was κ = 0.712 (*p* < 0.001) and κ = 0.481 (*p* < 0.001) when compared to FSA and H&E FFPE, respectively. Moreover, sensitivity and specificity have comparable results to already published series. In the tailored surgery era, it is mandatory to offer patients the treatment that best fits their conditions. FCM perfectly places in this context; thanks to its rapidity in specimen preparation and image acquisition, it may be possible to adapt the surgical strategy in real-time. During RC, to guarantee optimal oncological results, the opportunity to perform a sex-sparing technique in young patients motivated to preserve their function mostly depends on surgical margin negativity. Likewise, an intraoperative positive urethral margin may compromise the desire of patients to get a continent orthotopic neobladder regardless of the preoperative surgical planning. Continent urinary diversions, specifically orthotopic neobladders, currently represent patients' preferred option, as they offer improved aesthetics and quality of life (QoL) compared to other incontinent derivations. Establishing in real time that the tumor is confined to the bladder without infiltrating a neighbor's organs with reliability and remarkable quality could improve daily practice in ensuring patients the optimal surgical option.

Analogous to previously reported studies, the likelihood of BCa recurrence in our cohort was negligible (2.2%) [4]. Despite a negative margin status on FSA may theoretically decrease the onset of BCa recurrence after RC, a wide number of urologists remain confident of performing intraoperative FSA of the urethra and distal ureters during RC [11]. According to our findings, a negative pathology status identified at the time of FCM analysis was associated with a lower risk of BCa recurrence if we except only one case of negative ureteric FCM instead of a positive FSA. We cannot exclude that the low rate of local BCa recurrences observed could be attributable to the strategy of repeated ureteral sectioning in patients with adverse pathologic features rather than to the high NPV showed by either FCM or FSA methodology. Furthermore, due to the low number of UTUC recurrence observed as the competing risk of BCa potential metastatic disease (28.3%) reported in our dataset, we could not adjust our analysis for other potential confounders of the relationship between positive FCM and BCa recurrences. Nonetheless, the high values of sensitivity and specificity showed by FCM raise a question not only on the potential diagnostic role of this novel technology but also the undeniable potential clinical risk associated with the omission of intraoperative surgical margins evaluation.

If on one hand the sequential sectioning of the ureter to achieve a reliable negative margin status may explain the low incidence of ureteral BCa recurrences observed in our series, on the other hand, it does not avoid the eventuality of a synchronous or concealed UTUC, due to the intrinsic panurothelial nature of UC. Despite the lack of general consensus on performing urethral and ureteral sections intraoperatively, traditional FSA remains a time-consuming procedure with intrinsic risks of either potential loss or distortion of the specimen retrieved. Moreover, the process for the acquisition of FSA glass slides may be misleading by accidentally clearing the carcinoma during processing, consequently determining a false negative margin at the final evaluation [16]. Furthermore, the controversial role of the FSA is supported by the lack of strong recommendations in the current European guidelines [1]. Considering the intrinsic unreliability of positive FSA performed on a restricted number of samples, both surgeons and pathologists should consider the rationale of multiple intraoperative sampling during RC to obviate the rate of intraoperative false-negative margins and quick specimen deterioration before the definitive histologic analysis. According to our experience, the main advantages of FCM in the RC setting are represented by tissue preservation as the low discrepancy between ex vivo real-time examination and final specimen analysis.

In recent years, several authors suggested restricting intraoperative pathologic analysis only to patients affected by CIS [17,18]. However, the false-negative rate of FCM analysis in our series was relatively low and not consistent with the presence of CIS. Our data may underestimate CIS occurrence due to the small cohort and the retrospective analysis. Consequently, we strongly believe that real-time surgical specimen analysis can be considered a compelling crossroad during RC, especially in patients harboring CIS. However, even if we had respected this strategy, we would not have obtained a significant decrease in the false-negative rate in our series. Although our data may underestimate the real CIS occurrence due to the small cohort considered, we assume that FCM may still be considered a compelling alternative to FSA, even in patients harboring a concomitant CIS. Notably, the low false-negative rate and the ability to safely and quickly exclude tumor invasion of either the ureter or urethra represent the most relevant nuances for considering FCM as a reliable diagnostic test in the RC scenario. Despite the operating characteristics of FCM, there is undoubtedly a correlation between UTUC and FCM analysis that remains positive. However, the power of this association is unknown since we could not adjust our analysis for other potential confounders of this relationship due to the low number of UTUC recurrence (2.2%) and the competing risk of bladder primary potential metastatic disease (28.3%) reported in our dataset.

To increase overall survival (OS), 13% of patients (6/46) enrolled underwent NAC [1,19,20]. Due to the downstaging intent of NAC itself, it is reasonable to assume that surgical margin rates in our cohort may be partially underestimated. However, a recent

study [21] reported that FSA performance does not change in patients who underwent NAC, while in non-NAC patients, FSA detected a higher rate of positive ureteral margins, in particular, if they had a concomitant CIS in TURB specimens ($p = 0.033$). No correlation was observed between preoperative clinical factors and FSA positivity in the NAC cohort. Among patients who underwent NAC, only one patient (1/6, 16.7%) showed positivity in all urothelial surgical margins with perfect concordance between FCM, FSA, and permanent sections. He experienced bone metastases a few months after surgery with an MFS of 4 months. In a similar scenario, intraoperative analysis of surgical margins should be highly recommended, irrespective of NAC administration. Whether FCM might have limited benefit in guiding the postoperative surveillance of patients and influence oncological outcomes over other established risk factors remains yet to be determined. This is consistent with the absence of an association between adverse FCM and local recurrence, as well as overall mortality. However, it must be pointed out that our population is relatively small, with a short-term follow-up to draw significant conclusions about any survival outcome.

It should be emphasized that while FCM may give insight into the extent of tumor invasion, its primary use remains for intraoperative critical decision-making. Theoretically, FCM methodology could guide risk-adapted surveillance after surgery or real-time tailoring of the choice for urinary diversion. Finally, several low-volume centers may lack a highly-specialized pathologist or technician, precluding the possibility of performing in all patients an intraoperative evaluation of urethral and ureteral margins at the time of RC. In this setting, digitalizing pathological images may represent a valuable solution.

Our study is not devoid of limitations. First, the retrospective design and the relatively small series considered may result in a potential selection bias due to the surgeon's selection of anatomical specimens retrieved for either FCM or FSA. Moreover, the heterogeneity of the patient population in terms of inclusion criteria and clinicopathologic features may have jeopardized the interpretation of FCM and FSA outcomes. Additionally, we were unable to either perform subgroup analyses according to different FCM results (e.g., presence or absence of CIS) or to identify any significant predictor of survival outcomes in our cohort due to limited follow-up. Our dataset captured the frozen sections of the distal ureteric margins as the final, permanent section interpretation of the same residual tissue. As such, we could not delineate any skip lesions for ureters in which the final, proximal permanent margin is abnormal, despite a benign distal margin. As UTUC recurrence remains relatively low [4], an adequately powered randomized controlled trial remains challenging. Thus, the present study using our institutional database, including FCM and ureteric FSA with permanent pathology, for all patients may provide important insights in the near future. Therefore, prospective, larger multicentric studies with a centralized review of pathologic slides would be strongly advisable to confirm the reliability of the current results obtained.

Notwithstanding these limitations, the present study promotes the debate on the utility of routine FSA and the consideration of alternative methodologies for evaluating either distal ureteral or urethral margins at the time of RC. Routine FSA of the distal ureteric margin has poor sensitivity, which is marginally improved in patients with preoperative or concomitant CIS [17]. The predictive ability of intraoperative methodologies for upper tract recurrence remains questionable as several series have shown that permanent evaluation of ureteric margins of the proximal resection may be sufficient to predict potential upper tract recurrence and guide risk-adapted surveillance. Nonetheless, the process of sequential sectioning may remain valuable in preventing local recurrences, but it still requires additional prospective evaluation.

Despite data regarding the experience of pathologists performing FCM and FSA and its association with the final margin being limited, we firmly believe that introducing a standardized pathologic reporting scheme may increase the quality of RC specimen analysis pathologists experience. Additionally, the progressive adoption of digitalized histological imaging with novel artificial intelligence might significantly change the landscape of both surgical and pathologic RC workflow, consequently improving the diagnostic and predictive abilities of current prognostic models for UC.

## 5. Conclusions

FCM turned out to be a fast, safe, feasible, and reproducible method for the intraoperative assessment of urethral and ureteral surgical margins during ORC. Our study is the first to explore the application and feasibility of FCM on both urethral and ureteral margins in the RC setting. Compared to standard FSA, intraoperative FCM demonstrated adequate diagnostic performance and reproducibility in detecting urethral and ureteral malignant involvement at the time of RC for BCa. Our findings support the use of FCM for both urethral and ureteral margins during RC until cost-effectiveness studies can reliably identify patients who are unlikely to benefit from it.

**Author Contributions:** Conceptualization, R.P., R.M.S. and A.C.; methodology, A.C., C.T., R.P. and F.P.; validation, R.P., A.C. and R.M.S.; formal analysis, F.P. and U.A.; investigation, S.M.R. and M.V.; resources, R.P., A.C. and R.M.S.; data curation, A.I., A.R., C.T., S.M.R., M.V., F.P., S.M.P. and F.E.; writing—original draft preparation, F.P. and U.A.; writing—review and editing, R.P. and G.S.; visualization, R.P., A.C., G.S. and R.M.S.; supervision, R.P., A.C., G.S. and R.M.S. All authors have read and agreed to the published version of the manuscript.

**Funding:** This research received no external funding.

**Institutional Review Board Statement:** The study was conducted in accordance with the Declaration of Helsinki and approved by the Institutional Review Board (or Ethics Committee) of Campus Bio-Medico University (Number: FCM2020).

**Informed Consent Statement:** Informed consent was obtained from all subjects involved in the study. Written informed consent has been obtained from the patients to publish this paper.

**Data Availability Statement:** The data presented in this study is available on request from the corresponding author.

**Conflicts of Interest:** The authors declare no conflict of interest.

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
