# Peer review of "Real-Time Urethral and Ureteral Assessment during Radical Cystectomy Using Ex-Vivo Optical Imaging: A Novel Technique for the Evaluation of Fresh Unfixed Surgical Margins"

_curroncol, doi:10.3390/curroncol30030259_

Round 1
Reviewer 1 Report (Previous Reviewer 3)
Changes sufficient!
Author Response
We thank the reviewer for agreeing to review our paper and for the comment.
Reviewer 2 Report (New Reviewer)
Dear Authors,
I reviewed the study titled "Real time urethral and ureteral assessment during radical cystectomy using ex-vivo optical imaging: a novel technique for the evaluation of fresh unfixed surgical margins". It is a well-prepared study in general, but please pay attention to the corrections english spell checked and upload the study to the system again according to the journal rules. Thanks for your application as manuscript.
Yours sincerely
Author Response
We thank the reviewer for agreeing to review our paper and for the comments.
An extensive English review has been performed and the manuscript was reviewed by a native English speaker.
The study was uploaded to the system again according to journal rules.
Reviewer 3 Report (New Reviewer)
It is an interesting retrospective study aiming to evaluate the accuracy of fluoroscopic confocal microscopy for the detection of positive surgical margins.
Introduction: The issue of the need for intraoperative evaluation of surgical margins was adequately described. The necessity for the integration of this new technology in clinical practice was also stated. The purpose of this study was clearly described.
Materials and Methods: The study design was described in detail. Regarding the tissue preparation, the same specimen was referred from FCM to FSA? Was the normal saline wash performed in a specific way to avoid contamination of acridine orange? A comparison of the accordance between the FCM/H&E and FSA/H&E could be beneficial as the FCM could replace the FSA application.
Results: The results were adequately interpreted and the tables are very useful and comprehensive. On lines 153-156, the results are confusing. Did the FCM detect a different number of positive margins when compared with FSA and final histology? The language should be revised on these lines.
Discussion: All the main points of the study were discussed in comparison with the available literature.
Conclusion: The conclusion is in accordance with the results.
Author Response
Reply to the reviewer’s comment to curroncol-2198607
We thank again the Reviewer for comments, questions and suggestions to our manuscript.
They all were red and answered; when modifications of the manuscript were required, the modified text was included in the present letter and highlighted.
Reviewer #3
It is an interesting retrospective study aiming to evaluate the accuracy of fluoroscopic confocal microscopy for the detection of positive surgical margins. Introduction: The issue of the need for intraoperative evaluation of surgical margins was adequately described. The necessity for the integration of this new technology in clinical practice was also stated. The purpose of this study was clearly described.
We thank the reviewer for agreeing to review our manuscript and for the comments.
Materials and Methods: The study design was described in detail. Regarding the tissue preparation, the same specimen was referred from FCM to FSA?
We thank the reviewer for the suggestion.
The same specimen was referred for both FCM and FSA.
Lines 118-119: “After this processing, the same fresh unfixed specimens were removed from Vivascope and sent to standard FSA”
Was the normal saline wash performed in a specific way to avoid contamination of acridine orange?
We thank the reviewer for the comment.
After being treated with acridine orange, the specimen was fully washed with sterile saline solution. There was no need to perform saline was in a specific way to avoid contamination of acridine orange. Moreover, this procedure is easily reproducible as not requires a specific training.
A comparison of the accordance between the FCM/H&E and FSA/H&E could be beneficial as the FCM could replace the FSA application.
We thank the reviewer for his/her suggestion.
We performed an agreement analysis between FCM and FSA or H&E without comparing ex-vivo techniques to final H&E pathology because of intrinsic issue of FSA: the process for the acquisition of FSA glass slides could sometimes lead to the carcinoma being cleared from sections during sampling and consequently to a negative margin at final pathology, with a substantial number of false-negatives. Due to this concern, we were not able to provide reliable comparison data of FCM/H&E and FSA/H&E, thus we avoided the comparison suggested by the reviewer.
Lines 196-200: “Moreover, the process for the acquisition of FSA glass slides could sometimes lead to the carcinoma being cleared from sections during sampling and consequently to a negative margin at final pathology, with a substantial number of false-negatives. This is the reason why we avoided a direct comparison between FCM/FSA and H&E.”
Results: The results were adequately interpreted and the tables are very useful and comprehensive.
We thank the reviewer for agreeing to review our paper.
On lines 153-156, the results are confusing. Did the FCM detect a different number of positive margins when compared with FSA and final histology?
We thank the reviewer for the comment.
The section has been improved as suggested by the reviewer, showing the difference in rate of positive margins revealed by FCM in comparison to FSA and H&E for urethral and ureteral specimens separately. Tables related to these results were also improved accordingly.
Lines 153-156: “Compared to FSA, FCM reported a prevalence of positive surgical margins of 5/5 (100%) and 7/9 (77.8%) for urethral and ureteral specimens, respectively. While compared to final histology, FCM reported a prevalence of 5/6 (83.3%) and 7/9 (77.8%) positive surgical margins for urethral and ureteral specimens, respectively.”
Table 2. Concordance evaluation of FCM at urethral level between FSA and final histology.
Variable |
FSA |
Final Histology |
Prevalence |
5/5 (100%) |
5/6 (83.3%) |
Sensitivity |
80% |
66,7% |
Specificity |
97,8% |
97,5% |
Positive Predictive Value (PPV) |
80% |
80% |
Negative Predictive Value (NPV) |
97,6% |
95,1% |
K Cohen |
0,776 (p<0.001) |
0,691 (p<0.001) |
* Abbreviations: FCM: fluorescent confocal microscopy; FSA: frozen section analysis
Table 3. Concordance evaluation of FCM at ureteral level between FSA and final histology.
Variable |
FSA |
Final Histology |
Prevalence |
7/9 (77.8%) |
7/9 (77.8%) |
Sensitivity |
69,2% |
53,8% |
Specificity |
97% |
90,9% |
Positive Predictive Value (PPV) |
90% |
70% |
Negative Predictive Value (NPV) |
88,9% |
83,3% |
K Cohen |
0.712 (p<0.001) |
0.481 (p<0.001) |
The language should be revised on these lines.
We thank the reviewer for the suggestion. An extensive English review has been performed and the manuscript was reviewed by a native English speaker.
Discussion: All the main points of the study were discussed in comparison with the available literature. Conclusion: The conclusion is in accordance with the results.
We thank the reviewer for agreeing to review our paper as for the precious comments that allowed us to improve the manuscript.

This manuscript is a resubmission of an earlier submission. The following is a list of the peer review reports and author responses from that submission.
Round 1
Reviewer 1 Report
Please, describe comprehensively the studied method- what we know about it, who does it (pathologist?), learning curve, intra/inter individual variability, costs etc
The advantage over the widely used method (FSA) must be proved.
Demonstrate the efficacy of FSA (compared to a definitive FFPE)
Do the same with FCM. And compare the results.
Results
do not repeate the variables presented in the Table 1.
Present the clinical results (number of positive margins in FSA, FCM and FFPE. Histology (not hystology..), number of recurrences (local, urethral, upper tract)
OS, CCS, MFS - how many years? 1 year??
Table 2+3 - how many ureters were examined? 46 or 92?
Concordance with FSA was substantial/moderate, concordance with FFPE was worse. What is the concordance of FSA and FFPE?
Disscusion
You should discuss the results that were obtained. This discussion is a total mess. Repeatedly, numbers absolutely not presented in the results are discussed. The study does not say anything about the clinical impact of FCM on UC recurrence, etc. All are just speculations.
This is just a pilot retrospective study telling us that FSA in open RC is feasible and maybe comparable with FSA and thats it (Results of FSA itself were not presented). Is FCM better or worse than FSA and why?
Lmitations are too many. Try to reduce them.
Conclusions
Made on a basis of virtual results or hypotheses
Author Response
Please, describe comprehensively the studied method- what we know about it, who does it (pathologist?), learning curve, intra/inter individual variability, costs etc
We thank the reviewer for agreeing to review our paper and for the comment.
Tissue preparation has a dedicated section in the methods, nevertheless a paragraph regarding the fluorescent confocal microscopy (FCM) and how It works has been added to the manuscript.
FCM learning curve (LC) in the urology field has been described by Bertoni L. et al [1] about prostate and prostatic tissues as a short-term LC, while up to date this is the first study regarding FCM application for urethral and ureteral margins. The LC of FCM depends on tissue preparation (see the dedicated section in the manuscript) that only requires acridine orange and saline solution. After tissue preparation, the VivaScope translates the acquired image information from both lasers’ signals into pseudo-colored images that resemble hematoxylin and eosin (H&E) stain of conventional histological sections. Thus, the LC for image interpretation can be considered the same for conventional histopathological sections.
In this study we aimed to assess the diagnostic accuracy of FCM for ureteral and urethral margins before considering intra/inter individual variability, being this a novel technology that has never been explored in the field of urology except for prostate cancer and prostate tissue. In future perspective studies, this could be a secondary endpoint as to assess pathologists inter-reader variability.
Regarding costs, as explained in the methods section, the only material needed to prepare tissues are acridine orange and saline solution without the need of additional equipment except for the confocal laser. Moreover, in addition to conventional histological sections process, FCM allows to preserve tissue from potential distortion or loss avoiding the need of additional ureter resection.
[1] Bertoni, Laura et al. “Ex vivo fluorescence confocal microscopy: prostatic and periprostatic tissues atlas and evaluation of the learning curve.” Virchows Archiv : an international journal of pathology vol. 476,4 (2020): 511-520. doi:10.1007/s00428-019-02738-y
The advantage over the widely used method (FSA) must be proved. Demonstrate the efficacy of FSA (compared to a definitive FFPE). Do the same with FCM. And compare the results.
We thank the reviewer for the comment.
The aim of our study was to assess the feasibility and the reproducibility of FCM real-time assessment of urethral and ureteral margins and to compare with standard frozen section analysis (FSA). This was the first study exploring FCM accuracy in bladder cancer and thus our research question was explorative instead of confirmative. Further perspective studies and an external validation of our results are needed to prove the advantages over the standard FSA. Nevertheless, we think that this study could lay the groundwork for the development of a new technique able to overcome FSA intrinsic limitations.
We think that demonstrating the efficacy of FSA (compared to a definitive FFPE) could result out of the scope of our work. Moreover, as stated in the manuscript introduction, current international guidelines do not recommend to perform intraoperative FSA of ureteral margins during RC due to the controversial prognostic role and accuracy (page 2, lines 44-48).
The efficacy of the FCM could be assessed only comparing its results to FSA and final histology (considered the actual gold standard), and these information were already provided in the results section and displayed as tables (2 and 3 for urethral and ureteral margins, respectively).
Results
do not repeate the variables presented in the Table 1.
Present the clinical results (number of positive margins in FSA, FCM and FFPE. Histology (not hystology..), number of recurrences (local, urethral, upper tract)
We thank the reviewer for the remarkable suggestion.
We deleted repeated information in the manuscript and presented the clinical results as suggested by the reviewer. The manuscript and table 1 were improved accordingly.
OS, CCS, MFS - how many years? 1 year??
We thank the reviewer for the comment.
OS, CCS and MFS were calculated at the last date of patients’ follow-up that was (median) 11.5 months (IQR, 11-15 months). So, median follow-up was approximately one year.
Table 2+3 - how many ureters were examined? 46 or 92?
We thank the reviewer for the comment.
A total of 92 ureteral specimens were evaluated (46 right and 46 left ureteral margins). However, we considered right and left as the “same” margin (if both negative then were considered negative, if both positive then positive, and if only one was positive the whole was considered as positive surgical margin) in order to avoid information biases. Splitting ureteral side would not give us additional information, being the aim of the study to assess the concordance between FCM and conventional pathology.
Concordance with FSA was substantial/moderate, concordance with FFPE was worse. What is the concordance of FSA and FFPE?
We thank the reviewer for the punctual question.
Concordance between FSA and FFPE was slightly better than between FSA and FFPE, nonetheless adding this information to the manuscript could be in our thought out of scope. Furthermore, as already stated, current international guidelines do not recommend to perform intraoperative FSA of ureteral margins during RC due to the controversial prognostic role and accuracy (page 2, lines 44-48).
Disscusion
You should discuss the results that were obtained. This discussion is a total mess. Repeatedly, numbers absolutely not presented in the results are discussed. The study does not say anything about the clinical impact of FCM on UC recurrence, etc. All are just speculations.
We thank the reviewer for the comment.
In the discussion section we limited to discuss only results obtained. The numbers defined as “absolutely not presented” are instead derived number: i.e. BCa potential metastatic disease (28.3%) (page 7, line 215) was obtained by the difference of 100% and MFS 71.7% (see table 1). Other numbers were corrected accordingly and discussion section improved.
Nonetheless, considering the median follow-up of only 11.5 months, talking about clinical impact of FCM on UC recurrence would just be a speculation and this would go out of the scope of our work that aimed to assess the feasibility and the diagnostic accuracy of FCM.
This is just a pilot retrospective study telling us that FSA in open RC is feasible and maybe comparable with FSA and thats it (Results of FSA itself were not presented). Is FCM better or worse than FSA and why?
We thank the reviewer for the comment.
According to our findings, a negative pathology status identified at the time of FCM analysis was associated to a lower risk of BCa recurrence if we except only one case of negative ureteric FCM instead of a positive FSA.
The aim of our study was not to assess if FCM is better than FSA but to assess if they are comparable in terms of cancer infiltration detection. The advantages of FCM over FSA are the easier and faster preparation of tissue samples, the possibility to share the image acquired with other pathologists, and the ability to preserve tissue from distortion or potential loss.
Lmitations are too many. Try to reduce them.
We thank the reviewer for the suggestion.
Limitations were reduced and manuscript improved accordingly.
Conclusions
Made on a basis of virtual results or hypotheses
We thank the reviewer for the suggestion.
The conclusions were modified according to results and manuscript improved.
Reviewer 2 Report
The authors investigated the feasibility and the reproducibility of 16 fluorescent confocal microscopy (FCM) real-time assessments of urethral and ureteral margins during open radical cystectomy (ORC) for bladder cancer (BCa).
#1 The technical procedure of FCM remains unclear. Fluorescence is supposed to mean fluorescent staining with an antibody against a specific molecule, but it is unclear what antibody was used. Also, there is no fluorescently stained figure. Therefore, the meaning of FCM is unclear. Please explain it.
#2 The authors showed a high diagnostic agreement of urethral/ureteral margin for FCM, FSA, and H&E FFPE. To demonstrate the clinical superiority of FCM, the authors need to compare FCM and FSA in terms of "time and operational burden". In clinical practice, FSA is reported within 30 minutes of sample submission and it is not operationally troublesome. If the diagnostic accuracy is equivalent, the authors need to show the benefit of FCM.
#3 To address the utility of FCM, the authors need to show recurrence-free survival after surgery.
Author Response
The authors investigated the feasibility and the reproducibility of 16 fluorescent confocal microscopy (FCM) real-time assessments of urethral and ureteral margins during open radical cystectomy (ORC) for bladder cancer (BCa).
We thank the reviewer for agreeing to review our paper and for his/her comments.
#1 The technical procedure of FCM remains unclear. Fluorescence is supposed to mean fluorescent staining with an antibody against a specific molecule, but it is unclear what antibody was used. Also, there is no fluorescently stained figure. Therefore, the meaning of FCM is unclear. Please explain it.
We thank the reviewer for the comment.
#2 The authors showed a high diagnostic agreement of urethral/ureteral margin for FCM, FSA, and H&E FFPE. To demonstrate the clinical superiority of FCM, the authors need to compare FCM and FSA in terms of "time and operational burden". In clinical practice, FSA is reported within 30 minutes of sample submission and it is not operationally troublesome. If the diagnostic accuracy is equivalent, the authors need to show the benefit of FCM.
We thank the reviewer for the remarkable suggestion.
As stated in the methods section, “Tissue preparation, image acquisition and final evaluation required less than 5 minutes for each case” (page 3, lines 115-117) being significantly lower than the reported 30 minutes for FSA. Nonetheless, the aim of our study was to assess the feasibility and the diagnostic accuracy of FCM compared to FSA before exploring the time benefit derived from this diagnostic tool. Further perspective studies are needed to demonstrate the superiority in time of FCM over FSA.
However, the benefits of FCM over FSA are the easier and faster preparation of tissue samples (without the need of a technical expertise), the possibility to share the image acquired with other pathologists, and the ability to preserve tissue from distortion or potential loss. These advantages were displayed in the discussion section.
#3 To address the utility of FCM, the authors need to show recurrence-free survival after surgery
We thank the reviewer for the comment.
In our series, only one patient (2.2%) experienced an UC recurrence. Thus the RFS is of 97.8% (see table 1 CCS).
Reviewer 3 Report
Prata et al compare standard frozen section analysis of ureters and urethra at time of cystectomy with fluorescent confocal microscopy. Reported results indicate a high level of concordance between the two methods.
Cohort is arguably small but methodology is sound and presentation clear.
Major comments:
Where there a selection of cases included? It is implied that the 62 cases represent all cases during the time-frame but at the same time the main limitation stated is the selection bias. Is this selection bias strictly the surgeons choice of specimen. If so please reconsider phrasing since "selection-bias" normally is reserved for biased inclusion of research persons in the study.
Minor comments
Line 36: Indications "for"
Line 46-47: Ambiguous - please consider rephrasing
Line 251: In the method section it is described that the institutions Bca dataset was queried. It thus seems that the cohort is defined by the exposure (FCA) and not by the outcome hence it is arguably not a retrospective investigation.
Author Response
Prata et al compare standard frozen section analysis of ureters and urethra at time of cystectomy with fluorescent confocal microscopy. Reported results indicate a high level of concordance between the two methods. Cohort is arguably small but methodology is sound and presentation clear.
We thank the reviewer for agreeing to review our paper.
We acknowledge the relatively small cohort of our study, nonetheless this was the first study evaluating the application of FCM to urethral and ureteral margins at the time of cystectomy. Future perspective studies will need a larger sample size.
Major comments:
Where there a selection of cases included? It is implied that the 62 cases represent all cases during the time-frame but at the same time the main limitation stated is the selection bias. Is this selection bias strictly the surgeons choice of specimen. If so please reconsider phrasing since "selection-bias" normally is reserved for biased inclusion of research persons in the study.
We thank the reviewer for the question.
We presented the application of FCM to clinical practice. In order to produce more generalizable results, we avoided to select cases before the use of FCM, thus all patients who underwent RC in the timeframe considered underwent FCM evaluation of urethral and ureteral margins at the time of surgery. The surgeon did not perform any kind of specimen choice.
Minor comments
Line 36: Indications "for"
Line 46-47: Ambiguous - please consider rephrasing
We thank the reviewer for the comment.
Manuscript was improved accordingly.
Line 251: In the method section it is described that the institutions Bca dataset was queried. It thus seems that the cohort is defined by the exposure (FCA) and not by the outcome hence it is arguably not a retrospective investigation.
We thank the reviewer for the comment.
There was no specimen selection. “surgeon selection of anatomical specimens retrieved for either FCM or FSA”, we meant that at the time of RC is the surgeon that harvest the specimen from the bladder tissue, so there could be a potential selection bias due to the choice of a more or less suspicious margin. Nevertheless, this won’t influence FCM performance and accuracy, neither FSA or FFPE. In order to avoid confusion, that sentence was removed from the manuscript.